# Factors Influencing Occupational Health and Safety among Nursing Students in Their Clinical Placements: A Cross-Sectional Study

**DOI:** 10.3390/healthcare12090884

**Published:** 2024-04-24

**Authors:** Abdullah Suwailem Alsharari, Ali Kerari

**Affiliations:** 1Nursing Educator, Tubarjal Hospital, Tubarjal 74766, Aljouf, Saudi Arabia; aalshrari@moh.gov.sa; 2Nursing Administration and Education Department, College of Nursing, King Saud University, Riyadh 11421, Saudi Arabia

**Keywords:** occupational health, occupational safety, industrial health, nursing, nursing students

## Abstract

Occupational health and safety aims to promote and maintain optimal physical, mental, and social health for workers in their occupations. Within Saudi Arabia, adequate information must be gathered to address the various factors influencing occupational health and safety among nursing students to minimize occupational health hazards and ensure a safe clinical environment. This cross-sectional study was conducted in Riyadh, Saudi Arabia, between April and September 2023, involving 150 nursing students. Data collection included questions to gather sociodemographic information, and contained an instrument assessing the participants’ knowledge of occupational health and safety and evaluation of risk control in clinical environments. Blood and other bodily fluids, workplace violence, needle-stick injuries, and injuries caused by sharp instrument tools were identified as the most prevalent occupational risks among the participants. Most nursing students were aware of occupational health and safety, with a high level of compliance with occupational health and safety measures and personal protective equipment use. We also identified a statistically significant correlation between occupational health and safety knowledge and risk control evaluation among nursing students. To ensure a safe and beneficial clinical training area, student nurses must complete extensive occupational health and safety courses before moving to clinical areas to reduce potential hazards that may affect their lives.

## 1. Introduction

Nursing students are considered a high-risk group because of their lack of experience in dealing with workplace hazards and complex environments, along with a shortage of instructors, which may negatively affect their internship pathways [1]. Because of their direct contact with patients, inadequate training in safety culture, and readiness to take risks, nursing students are likely to be exposed to pathogens and occupational health and safety (OHS) risks during their internships. Moreover, nursing students’ low OHS knowledge could increase their exposure to numerous hazards and high-risk work environments, ultimately threatening their lives [2]. Therefore, increasing nursing students’ knowledge and awareness of OHS, as well as that of their trainers, can promote safe practices and minimize their exposure to hazards and risky environments [3]. This can be achieved by ensuring that these individuals possess comprehensive knowledge about the risk factors of OHS such as physical, mental, and social issues that impact safety. To acquire such knowledge, nursing students must have increased awareness of the barriers to achieving optimal OHS in the workplace.

Several studies reported that more than half of the students were completely aware of personal protective equipment (PPE) and safety laws; the most frequently mentioned health issues in these studies were anxiety, musculoskeletal pain, and lower back discomfort [4,5,6]. Similarly, Qaraman et al. found that 21% of nursing students had experienced one or more needle-stick injuries (NSIs) [7]. Another study observed that a lack of understanding of infarction control techniques and a lack of commitment to immunization protocols were highly associated with an elevated incidence of NSIs and sharps injuries [7]. Tariah et al. reported that nurses had a greater incidence of work-related musculoskeletal diseases than other health professionals [8]. In addition, 63.8% of the nurses in the study reported experiencing lower back pain in the previous 12 months. Therefore, raising nursing students’ awareness about the severe physical damage caused by work-related musculoskeletal problems is crucial.

According to Alshahrani et al., abuse and violence in the health field have become ubiquitous and serious problems; specifically, they reported that healthcare personnel are more prone to experiencing workplace violence today than they were in the past [9]. Al-Shamlan et al. discovered that roughly three in ten (30.7%) had experienced verbal abuse; moreover, most workers who experienced abuse had not reported it because previous complaints had resulted in negative consequences [10].

To date, only a few quantitative studies have assessed factors such as awareness and knowledge of OHS and exposure to occupational hazards among nursing students in Saudi Arabia; elucidating the effects of such factors could help to minimize occupational health hazards and identify aspects that contribute to safe practices and lower exposure to hazards and risky environments. Therefore, this study gathered information to address the various factors influencing OHS among nursing students. To address the lack of correlational descriptive studies on the above-mentioned factors, this study investigated nursing students’ awareness and knowledge of OHS and their exposure to occupational health hazards during their clinical placements at King Saud University in Saudi Arabia.

## 2. Materials and Methods

### 2.1. Study Design and Setting

A web-based cross-sectional study was conducted at King Saud University in Riyadh from April to September 2023. The study population comprised nursing students from King Saud University, with a total of 150 nursing students recruited for participation.

### 2.2. Participants

This study used convenience sampling to recruit nursing students performing their clinical rotations at King Khalid University Hospitals during their fourth through twelfth semesters of study to complete a questionnaire. This study employed convenience sampling to allow simple data collection and facilitate hypothesis development.

The sample size was determined through a statistical G power analysis. The significance criterion (α), the population effect size (the estimated size of the association between the independent variables and the dependent variables), and statistical power are essential elements of statistical inference. According to the guidelines for power analysis, the suitable sample size was estimated at 120. The final sample size was increased to 150 participants to account for 20% attrition.

A total of 150 students completed the survey. Males comprised 60% of the study population, and most participants were aged between 20 and 27 years old; 27% were older than 27. The year of study for students was categorized into two levels: lower level (first to second year of study, *n* = 79, 53%) and upper level (third to fourth year of study, *n* = 71 students, 47.3%) (Table 1).

### 2.3. Data Collection Procedures and Measures

Potential participants were approached at the College of Nursing according to their class schedules. The online survey link, which included an overview of the study, eligibility for participation, and informed consent, was shared through a barcode. Participants could respond from their smartphone, tablet, or laptop. The online self-administered questionnaire comprised two main sections. The first section gathered sociodemographic information, and the second section assessed participants’ knowledge of occupational health and safety and their evaluation of risk control in clinical environments. All research activities were closely monitored throughout the data collection phase; the lead investigator reviewed the quality of the data and the completeness of the questionnaires after each day of data collection. 

The questionnaire was a modified version of a previous questionnaire developed by Eyi et al. in which one item did not apply to Saudi Arabia [11]. The original questionnaire had 29 questions: 4 questions were related to students’ demographic characteristics, and 25 questions assessed their perspectives and knowledge regarding OHS [11]. The latter set of questions presented problems that students face in the clinical practice environment, with the answers categorized into four subcategories: identifying threats, assessment of OHS situations in nursing students, evaluation of risk control, and assessment. In this study, one of the questionnaire items from the original questionnaire was deleted because it was related to regulations and legislation specific to Turkey. 

Two subscales (awareness of the OHS situation and evaluation of risk control) were used, each of which consisted of several statements. Each statement was answered on a 4-point Likert scale ranging from 1 (completely disagree) to 4 (completely agree). These two subscales were used to assess students’ awareness of the OHS situation and their evaluation of risk control. Higher scores indicated higher knowledge levels of OHS and proper compliance with OHS measures. The Cronbach’s α values for the awareness of the OHS situation subscale and the evaluation of risk control subscale were 0.82 and 0.79, respectively. 

### 2.4. Data Analysis

All statistical analyses were performed with SPSS V.29. Descriptive statistics (frequencies, percentages, and means) were used to describe the sample variables. Independent samples *t*-tests were conducted to evaluate significant differences between groups. The chi-square test was used to examine the relationships between categorical variables, and the Pearson correlation coefficient was used to examine the relationship between continuous variables. All the assumptions related to these statistical tests were met. A probability value of *p* < 0.05 was considered statistically significant.

### 2.5. Ethical Approval

The study was conducted in accordance with the World Medical Association Declaration of Helsinki, and approved by the Institutional Review Board of the King Saud University (Ref #: KSU-HE-22-641 and dated 11 October 2022). Informed consent was obtained from all participants, who were provided with detailed information about the study. Additionally, participants were assured that their data would be treated with the utmost confidentiality and accessible only to the research team members.

## 3. Results

The findings were addressed within the framework of the following subcategories: identifying threats, assessment of OHS knowledge among student nurses, risk control evaluation, and nursing students’ perspectives on the origin of hazards, threat sources, and measures taken during OHS incidents.

### 3.1. Identifying Potential Threats

The questionnaire contained specific questions related to OHS threats in clinical practice. The first question was “What kind of dangerous situations related to OHS did you encounter in clinical practice?” Over half of the nursing students (60.7%) reported that contact with blood and body fluids (BBFs) was the largest threat. The second-largest threat was physical and verbal assault by the supervising nurse or doctor (52.7%), colleagues (52%), and patients or their relatives (52%). A total of 68 nursing students (45.3%) had an NSI after injection, and 43.3% of nursing students were exposed to injuries caused by sharps or drilling tools during their clinical practice. The differences in threats encountered between genders and academic levels were not statistically significant (*p* > 0.05) (see Table 2).

Regarding the identification of problems related to PPE, the most common problem was the lack of gowns and goggles, which was reported by 44.7% of nursing students. Furthermore, 38.7% of the participants reported using only one glove because of an insufficient supply, and 34.7% reported caring for many patients with the same gloves. In addition, we did not observe any statistically significant differences between academic levels or genders in recognizing PPE concerns (*p* > 0.05).

Regarding symptoms during clinical practice, most students reported suffering from headache (65.3%), stress (63.3%), exhaustion (54%), backache (56%, 84), early fatigue (54.7%), and anxiety or anger (50.7%). Participants reported that hand antiseptic (58%) and latex (36%) were the most utilized chemicals during clinical practice. A significant relationship was observed between gender (male and female) and the use of latex (*p* < 0.05); however, no significant relationship was detected between academic level or gender and chemical use (*p* > 0.05).

### 3.2. Assessment of OHS Knowledge

Table 3 presents the findings related to nursing students’ OHS awareness and knowledge. Most nursing students (67.3%) agreed that OHS was related to and impacted their careers, and 62% reported that they were knowledgeable about OHS specific to their profession. As shown in Table 3, a significant difference was found between academic levels regarding knowledge about OHS (*p* < 0.05). Moreover, 71.3% of nursing students indicated that they knew the definition of occupational diseases. Most of the students (69.3%) expressed that they preferred to have OHS classes in the curriculum. In addition, 66% stated that they were aware of OHS commitments and the rights of healthcare personnel and patients in hazardous conditions. Approximately half of the nursing students (51.3%) stated that they knew about OHS regulations and guidelines. No significant differences were observed in OHS knowledge between academic levels or genders (*p* > 0.05).

### 3.3. Risk Control Evaluation

As shown in Table 4, which provides information on risk control evaluation, 75.3% of the students reported that they showed the required sensitivity to OHS guidelines. Most students (76.7%) believed that their job required PPE, and most (76%) reported that they demonstrated appropriate sensitivity toward using PPE in training areas. On the other hand, 68.6% of students believed that OHS safeguards were adequate. As shown in Table 4, a significant difference was found between academic levels regarding the belief that the nursing profession requires PPE (*p* < 0.05). A significant difference in students’ risk control evaluation of OHS (sum score) was observed between academic levels (*p* < 0.01), whereas no statistically significant differences in the sum score of risk control evaluation were found between genders.

A moderate positive correlation was observed between OHS-related knowledge and the levels of risk control evaluation (r = 0.52, *p* < 0.001). Participants with adequate OHS-related knowledge reported higher levels of risk control evaluation.

### 3.4. Nursing Students’ Perceptions of Hazard Sources and Compliance with OHS Precautions

As shown in Table 5, the students experienced occupational hazards most often while administering routine care to patients (66.7%), monitoring vital signs in the clinic (64.4%), placing patients in beds or transporting them (63.8%), performing interventions in emergency services (61.3%), obtaining a patient’s medical history (57.7%), delivering patients’ laboratory samples (56.7%), administering therapy in the clinics (55.7%), preparing medicines for treatment (53.3%), and recapping needles (52%). No significant differences were observed in scenarios that exposed students to occupational hazards between academic levels or genders (*p* > 0.05).

Regarding nursing students’ perceptions of the reasons for occupational accidents, most reported long working hours (75.3%), followed by inexperience in dealing with hazards (71.3%), not using PPE (70%), and an intensive work pace and lack of attention (68.7%). A total of 64.7% of nursing students reported receiving rest leave when they encountered occupational dangers in training areas, and 69.3% of participants reported that their encounters with situations that threatened their OHS were documented. Most students (84%) stated that their health was their main priority in hazardous situations. 

## 4. Discussion

This study aimed to determine nursing students’ knowledge of OHS and their exposure to occupational health hazards during their clinical practice. The discussion of the findings focuses on the risk assessment process through the following subcategories: identifying threats, nursing students’ assessment of OHS knowledge, risk control evaluation, and perceptions of hazard sources and compliance with OHS precautions. 

### 4.1. Identifying Threats

Exposure to bodily fluids and blood was the most reported threat. Occupational blood and body fluid exposure (OBBE) is a substantial concern for nurses who work in fields that require them to regularly touch patients and their blood or other body fluids. Nurses have a high risk of acquiring bloodborne viruses, such as hepatitis B virus (HBV), hepatitis C virus (HCV), and human immunodeficiency virus (HIV) [4]. Similarly, Yasin et al. reported that healthcare workers had a higher risk of acquiring bloodborne viruses, such as HIV, HBV, and HCV, because of unintended exposure to BBFs [12]. This is among the most severe public health problems that healthcare workers face while they treat patients. Thus, internship and training center units at nursing colleges should focus on teaching students about preventative measures to mitigate OBBE risk. At clinical training sites, nursing students must be aware of infection control protocols, including PPE use (i.e., wearing gloves, masks, and goggles and safe needle handling practices). Accordingly, stakeholders and policymakers must increase nursing students’ knowledge by providing regular training and supervision to ensure that they adhere to safety precautions and procedures.

Our study found that the second-most reported threat was physical and verbal assault from supervising nurses and doctors, colleagues, and patients or their relatives. Di Prinzio et al. also found that workplace violence significantly affected employees’ health [13]. Workplace violence results in work overload, job dissatisfaction, fatigue, and exhaustion, which often lead to high rates of turnover and absenteeism. Violent assaults not only have psychological, physical, organizational, and professional repercussions but also decrease an organization’s performance. To prevent workplace violence, Di Prinzio et al. recommended that healthcare organizations develop targeted organizational policies and safety training programs for workplace violence management; implement effective communication training to facilitate earlier detection of potentially aggressive and violent behavior; establish reporting procedures; and provide medical, psychological, and legal support after violent incidents [13]. In addition, academic advising and counseling units should address all types of violence at clinical training sites. This would involve inviting and encouraging students to visit the counseling units to discuss and review their academic and non-academic issues. In addition, academic advisors can provide supportive academic advising by identifying personal and social problems at clinical sites. This would ultimately contribute to creating safe work environments for nursing students. 

NSIs and accidents caused by sharps or drilling tools are among the greatest threats healthcare employees face. Healthcare employees risk contracting life-threatening illnesses that are spread through the blood, such as HIV, HBV, and HCV. Our results showed that 45.3% of nursing students (*n* = 65) experienced an NSI and that 43.3% of students (*n* = 65) were injured by sharps or drilling tools during clinical practice. Needles, sharp instruments, and other devices that penetrate the skin are occupational hazards for healthcare personnel [14,15]. Therefore, healthcare organizations and nursing programs should ensure that students receive comprehensive education and training on proper techniques for needle disposal as well as safe work practices.

This study found that the most common problem that nursing students faced with PPE was an insufficient supply of gowns and goggles. This represents a threat to nursing students because it could increase their exposure to contaminants from bloodborne pathogens. Nurses frequently fail to wear all the required personal safety equipment, placing them at risk of contracting bloodborne diseases. Yasin et al. found that healthcare employees who did not utilize eye goggles had a higher risk of OBBE [12]. Healthcare institutions are responsible for monitoring and maintaining an adequate supply of PPE, including gowns and goggles. Nursing students and faculty should collaborate with institutional representatives to ensure that the PPE supply is sufficient, which would require an increase in their level of awareness about the importance of PPE use and its impact on student safety. 

Regarding symptoms experienced during clinical practice, the students experienced headaches, stress, exhaustion, early fatigue, and anxiety. These symptoms resulted in academic burnout and affected the OHS of the nursing students. Similarly, Hwang et al. reported that anxiety, stress, and depression affected academic burnout in students who had no clinical practice experience [16]. They also found that stress and satisfaction influenced academic burnout in nursing students who had clinical experience. Our results highlight the importance of creating and implementing intervention programs that reduce academic burnout in nursing students. To improve nursing students’ satisfaction with their majors and their values as nurses, clinical practice settings and programs must be re-evaluated and improved. Furthermore, schools must support students and coordinate with clinical site institutions to reduce stress, depression, and anxiety among nursing students and improve their satisfaction with their majors. 

Consistent with a study conducted by Eyi and Eyi [11], our study found a significant difference in chemical exposure during clinical practice between genders and reported that female students were more affected by latex allergies. Therefore, individuals who are at risk of developing an antiseptic or latex allergy during their education should be identified, and preventive measures should be taken to prevent life-threatening reactions such as anaphylaxis.

### 4.2. Assessment of OHS Knowledge

Our findings reveal a high level of OHS knowledge among nursing students at King Saud University, reflecting an advanced level of awareness and knowledge about the fundamentals of OHS and demonstrating the high quality of education at King Saud University. We did not observe any significant differences in OHS knowledge between academic levels and genders. Furthermore, the existing literature on OHS knowledge indicates that nursing students have a comprehensive understanding of OHS before advancing to clinical areas. Alharbi et al. found that the Nursing College at King Saud University offered high-quality education and a range of courses to meet Saudi Arabia’s need for nursing professionals [17]. Conversely, Yasin et al.’s study revealed that the proportion of student nurses who had received training in preventing occupational infections was low [12]. To meet the needs of a growing population, the nursing workforce must be skilled and well-educated. Therefore, universities should establish a strong system for teaching students how to be safe in clinical areas before moving to training areas.

Alharbi et al. reported that the high knowledge of OHS among nursing students at King Saud University signifies that they perceive the educational environment more positively than students at other universities [17]. In the current study, more than half of nursing students acknowledged the link between OHS and their profession. Consistent with our results, Eyi and Eyi’s study found that nearly all students (95.7%) were aware of OHS, and 52.8% of students identified OHS courses as a source of information [11]. To ensure adequate OHS knowledge, Ugur et al. indicated that nursing students should receive OHS training from an expert in the field before entering clinical rotations [18]. These studies suggest that the school’s curriculum is the primary source of OHS knowledge for nursing professionals.

### 4.3. Risk Control Evaluation 

In our risk control evaluation, nursing students showed a high level of knowledge of how to use PPE to reduce occupational incidents, pathogen transmission, and possible dangers in a wide range of occupations. Most students reported that they followed OHS rules as required; additionally, most stated that their jobs required PPE and that they used PPE in training areas with adequate care. Similarly, most students in Eyi and Eyi’s study reported that they were aware of the need for PPE in clinical settings, and almost all of them (97.1%) indicated that PPE was necessary in their professions [11]. 

According to Elshaer et al., student nurses’ opinions regarding PPE use were predominantly positive, indicating that they received education in this area. The authors found that nursing students utilized protective items at a relatively high rate [19]. We observed significant differences in risk control and evaluation between academic levels. In alignment with a study by Tuan et al., we observed a significant relationship between the duration of training, academic level, and nursing students’ compliance with risk control and evaluation [20]. Colet et al. reported a statistically significant difference in compliance with standard precautions between academic levels [21]. These data suggest that more clinical experience and academic years are related to higher compliance with standard precautions. Students in the bridge nursing program and those in later academic years have more clinical exposure, which is related to superior compliance with precautions. These results conflict with the findings of Eyi and Eyi, who observed no significant differences in compliance with OHS measures and PPE usage between academic levels [11].

Our findings revealed a statistically significant relationship between nursing students’ assessment of OHS situations and the level of risk control evaluation; specifically, as nursing students’ level of OHS knowledge increased, their risk control evaluation level also increased. This indicates that when nursing students have more knowledge about OHS, their level of risk control evaluation is higher. Olcay et al. found significant improvements in the students’ knowledge levels after an OHS course and concluded that all students in universities should receive OHS education [22]. Thus, increasing students’ knowledge and awareness of OHS would increase their understanding of risk management approaches to OHS, potentially providing students with a structured foundation for problem solving. This could benefit them in clinical reasoning regarding safety and health problems and strengthen their self-care.

### 4.4. Nursing Students’ Perceptions of Hazard Sources and Compliance with OHS Precautions 

The students identified different types of situations that potentially exposed them to occupational hazards in the clinic. Our study indicated that the nursing students were most often exposed to occupational risks when providing routine care to patients, performing interventions in emergency services, and recapping needles after use. Our findings are somewhat consistent with other studies, in which students stated that the most likely reasons for occupational injuries were a lack of PPE use, inexperience, and long working hours [11,19]. Nursing programs and healthcare facilities should foster a culture of workplace safety that ensures the well-being of nursing students. For example, nursing students are encouraged to seek help from experienced healthcare professionals who can provide supervision, guidance, and support during clinical practice. Clinical preceptors or mentors play major roles in addressing students’ concerns, offering feedback, and ensuring safe practices.

### 4.5. Implications for Occupational Health in Nursing Practice 

This research provides potential strategies for enhancing students’ OHS. The students reported that they were aware of the hazards associated with nursing. In college, first- and second-year nursing students should be familiar with OHS, which can improve their knowledge of clinical training risks, and they gain extensive hands-on experience in dealing with such risks by the end of their internship year. Employing risk evaluation represents a significant step in building students’ OHS knowledge. In addition, implementing courses related to manual patient handling may help students avoid injuries, such as back pain. Establishing an OHS curriculum that extends from the first academic year until graduation would provide nursing students with a comprehensive understanding of OHS; such a curriculum could involve pre-practical exams before students begin clinical practice. In clinical settings, clinical supervision plays an integral part in ensuring workplace safety for nursing students during their clinical practice. This involves monitoring and evaluating nursing students’ occupational health and safety practices, providing feedback, and identifying areas for improvement. 

### 4.6. Study Limitations 

This study’s cross-sectional design is a limitation of the study; this design makes it difficult to establish cause-and-effect associations because it only reflects a one-time assessment of the alleged cause and effect. The time of the cross-sectional snapshot may not reflect the group’s typical or overall behavior. Another limitation is the issues associated with online surveys, including internet access problems, response bias, survey fatigue, and a large number of unanswered questions.

## 5. Conclusions

This study investigated the awareness and knowledge of OHS and exposure to occupational health hazards among nursing students during their clinical placement and identified factors contributing to safe practices and reduced risk exposure. The findings revealed that the following occupational hazards were the most common: blood and other body fluids, workplace violence, and NSIs. We observed a statistically significant correlation between nursing students’ OHS knowledge and their risk control evaluation. To ensure a safe and beneficial clinical training area, student nurses must complete extensive OHS courses before moving to clinical areas and while working in clinical areas to reduce potential hazards that may affect their lives. Future studies should include other nursing schools and collect data on OHS. This would elucidate the diverse OHS experiences of nursing students regarding the risk assessment process from a wide range of educational perspectives.

## Figures and Tables

**Table 1 healthcare-12-00884-t001:** Demographic characteristics.

Variables	*n* (%)
Age	
Under 27	110 (73.3%)
27 or older	40 (26.7%)
Gender	
Male	90 (60%)
Female	60 (40%)
Year of study	
Lower academic level (first- and second-year nursing students)	79 (53%)
Higher academic level (third- and fourth-year nursing students)	71 (47%)
Training area	
Inpatient	92 (61%)
Outpatient	48 (32%)
Others	10 (7%)
Student type	
Full-time student	103 (68.67%)
Part-time student	47 (31.33%)
Marital status	
Single	119 (79%)
Married	31 (21%)

**Table 2 healthcare-12-00884-t002:** Nursing students’ experiences of occupational risks and hazards of OHS in clinical practices using the chi-square test.

	Gender			Academic Level			
	Male	Female		Lower Level	Upper Level		Total
Identifying Threats	*n* (%)	*n* (%)	*p* Value	*n* (%)	*n* (%)	*p*-Value	*n* (%)
Dangerous situations related to OHS							
Contact with blood and body fluids	55 (36.7%)	36 (24%)	0.89	50 (33%)	41 (27.3%)	0.48	91 (60.7%)
Physical/verbal assault by healthcare providers	43 (28.7%)	36 (24%)	0.14	46 (30.7%)	33 (22%)	0.15	79 (52.7%)
Needle-stick injury after injection	43 (28.7%)	25 (16.7%)	0.46	38 (25.3%)	30 (20%)	0.47	68 (45.3%)
Falling, slipping, strains, and dropping of materials	44 (29.3%)	27 (18%)	0.64	37 (24.7%)	34 (22.7%)	0.89	71 (47.3%)
Contact with chemical additives or liquids	36 (24.3%)	26 (17.6%)	0.76	34 (23%)	28 (18.9%)	0.65	62 (41.3%)
Physical/verbal assault by colleagues	48 (32.2%)	30 (20.1%)	0.63	43 (28.9%)	35 (23.5%)	0.47	78 (52%)
Sharp or drilling tool injury (e.g., scalpel and scissors)	44 (29.3%)	21 (14%)	0.09	40 (26.7%)	25 (16.7%)	0.05	65 (43.3%)
Physical/verbal assault from patients or their relatives	48 (32%)	30 (20%)	0.68	44 (29.3%)	34 (22.7%)	0.33	78 (52%)
PPE usage problems in clinical practice							
Having to use a single glove due to a lack of gloves	34 (22.7%)	24 (16%)	0.78	34 (22.7%)	24 (16%)	0.24	58 (38.7%)
Providing care to multiple patients with the same single glove	34 (22.7%)	18 (12%)	0.32	30 (20%)	22 (14.7%)	0.36	52 (34.7%)
Entering rooms of patients with respiratory contagious diseases without a mask due to no masks being available	34 (22.7%)	22 (14.7%)	0.89	30 (20%)	26 (17.3%)	0.86	56 (37.3%)
Not wearing gowns or goggles due to a lack of PPE resources	42 (28%)	25 (16.7%)	0.54	34 (22.7%)	33 (22%)	0.67	67 (44.7%)
Findings and symptoms in clinical practice							
Headache	55 (36.7%)	43 (28.7%)	0.18	54 (36%)	44 (29%)	0.41	98 (65.3%)
Early fatigue	46 (30.7%)	36 (24%)	0.28	42 (28%)	40 (26.7%)	0.69	82 (54.7%)
Exhaustion	46 (30.7%)	35 (23.3%)	0.38	41 (27.3%)	40 (26.7%)	0.58	81 (54%)
Burning eyes/throat	27 (18%)	18 (12%)	0.10	25 (16.7%)	20 (13.3%)	0.64	45 (30%)
Nosebleed	18 (12%)	19 (12.7%)	0.10	22 (14.7%)	15 (10%)	0.34	37 (24.7%)
Skin irritation	17 (11.3%)	18 (12%)	0.11	22 (14.7%)	13 (8.7%)	0.16	35 (23.3%)
Allergies	23 (15.3%)	20 (17.2%)	0.30	28 (18.7%)	15 (10%)	0.05	43 (28.7%)
Breathing difficulties	26 (17.3%)	17 (11.3%)	0.94	26 (17.3%)	17 (11.3%)	0.22	43 (28.7%)
Backache	48 (32%)	36 (24%)	0.42	41 (27.3%)	43 (28.7%)	0.28	84 (56%)
Anxiety or anger	42 (28%)	34 (22.7%)	0.23	38 (25.3%)	38 (25.3%)	0.50	76 (50.7%)
Stress	54 (36%)	41 (27.3%)	0.29	47 (31.3%)	48 (32%)	0.30	95 (63.3%)
Common areas of the body injured in clinical practice							
Head	22 (14.7%)	13 (8.7%)	0.69	18 (12%)	17 (11.3%)	0.86	35 (23.3%)
Forearm, wrist, palm, or finger	33 (22%)	27 (18%)	0.30	37 (24.7%)	23 (15.3%)	0.07	60 (40%)
Patella, calf, or foot	19 (12.7%)	13 (8.7%)	0.93	20 (13.3%)	12 (8%)	0.20	32 (21.3%)
Mental damage	18 (12%)	18 (12%)	0.16	20 (13.3%)	16 (10.7%)	0.69	36 (24%)
Chemicals contacted in clinical practice							
Hand antiseptic	47 (31.3%)	40 (26.7%)	0.07	46 (30.7%)	41 (27.3%)	0.95	87 (58%)
Formaldehyde, glutaraldehyde, ethylene oxide, and antineoplastic cancer drugs	17 (11.3%)	14 (9.3%)	0.51	17 (11.3%)	14 (9.3%)	0.78	31 (20.7%)
Latex	26 (17.3%)	28 (18.7%)	0.02 *	28 (18.7%)	26 (17.3%)	0.88	54 (36%)

Note: OHS: occupational health and safety; PPE: personal protective equipment; * *p* < 0.05.

**Table 3 healthcare-12-00884-t003:** The assessment of OHS knowledge and awareness, using the independent samples *t*-test and the chi-square analysis.

	Gender			Academic Level			
	Male	Female		Lower Level	Upper Level		Total
Nursing Students’ Awareness of OHS	Mean (SD)/*n* (%)	Mean (SD)/*n* (%)	*p*-Value	Mean (SD)/*n* (%)	Mean (SD)/*n* (%)	*p*-Value	Mean (SD)/*n* (%)
Thinking that OHS is related to my profession and affects it							
Yes	60 (40%)	41 (27.3%)	0.83	49 (32.7%)	52 (34.7%)	0.14	101 (67.3%)
Being knowledgeable about OHS							
Yes	53 (35.3%)	40 (26.7%)	0.33	42 (28%)	51 (34%)	0.01 *	93 (62%)
Knowing the definition of occupational disease							
Yes	60 (40%)	47 (31.3%)	0.12	54 (36%)	53 (35.3%)	0.39	107 (71.3%)
Requesting an OHS class in the curriculum							
Yes	58 (38.7)	46 (30.7%)	0.11	52 (34.7%)	52 (34.7%)	0.32	104 (69.3%)
Being aware of OHS commitments and the rights of patients and health workers in hazardous situations							
Yes	61 (40.7%)	38 (25.3%)	0.57	50 (33.3%)	49 (32.7%)	0.46	99 (66%)
Knowing OHS regulation novelties							
Yes	47 (31.3%)	30 (20%)	0.79	35 (23.3%)	42 (28%)	0.06	77 (51.3%)
Awareness of OHS ‘sum score’	15.81 (4.1)	16.50 (4.2)	0.32	15.46 (4.1)	16.79 (4.1)	0.05	150

Note: OHS: occupational health and safety; * *p* < 0.05.

**Table 4 healthcare-12-00884-t004:** Nursing students’ compliance with OHS measures and PPE usage, using the independent samples *t*-test and the chi-square analysis.

	Gender			Academic Level			
	Male	Female		Lower Level	Upper Level		Total
Evaluation of Risk Control	Mean (SD)/*n* (%)	Mean (SD)/*n* (%)	*p*-Value	Mean (SD)/*n* (%)	Mean (SD)/*n* (%)	*p*-Value	Mean (SD)/*n* (%)
Applying required OHS rules in clinical practice.							
Yes	68 (45.3%)	45 (30%)	0.93	56 (37.3%)	57 (38%)	0.18	113 (75.3%)
Thinking that the OHS measures taken are sufficient.							
Yes	59 (39.3%)	44 (29.3%)	0.31	50 (33.3%)	53 (35.3%)	0.13	103 (68.6%)
Applying required PPE rules in clinical practice.							
Yes	66 (44%)	48 (32%)	0.34	58 (38.7%)	56 (37.3%)	0.43	114 (76%)
Thinking that nursing profession requires PPE.							
Yes	69 (46%)	46 (30.7%)	0.90	55 (36.7%)	60 (40%)	0.03 *	115 (76.7)
Risk control evaluation of OHS ‘sum score’.	11.4 (2.9)	11.82 (2.7)	0.42	10.91 (2.8)	12.34 (2.6)	0.002 **	150

Note: OHS: occupational health and safety; PPE: personal protective equipment; * *p* < 0.05; ** *p* < 0.001.

**Table 5 healthcare-12-00884-t005:** Nursing students’ perceptions on hazard sources and compliance with OHS precautions using the chi-square analysis.

	Gender			Academic Level			
	Male	Female		Lower Level	Upper Level		Total
Assessment	Mean (SD)/*n* (%)	Mean (SD)/*n* (%)	*p*-Value	Mean (SD)/*n* (%)	Mean (SD)/*n* (%)	*p*-Value	Mean (SD)/*n* (%)
Practice made and hazardous situation exposed in clinic							
Providing regular care to patients	64 (42.7%)	36 (24%)	0.15	52 (34.7%)	48 (32%)	0.81	100 (66.7%)
Preparing drugs for treatment	52 (34.7%)	31 (20.7%)	0.46	41 (27.3%)	42 (28%)	0.37	83 (55.3%)
Conducting interventions in emergency services	56 (37.3%)	36 (24%)	0.78	50 (33.3%)	42 (28%)	0.60	92 (61.3%)
Applying treatments in the clinic	51 (34.2%)	32 (21.5%)	0.77	45 (30.2%)	38 (25.5%)	0.74	83 (55.7%)
Tracking vital signs in the clinic	58 (38.9)	38 (25.5%)	0.99	51 (34.2%)	45 (30.2%)	0.97	96 (64.4%)
Recapping needles	48 (32%)	30 (20%)	0.68	41 (27.3)	37 (24.7%)	0.97	78 (52%)
Taking a patient’s story	49 (23.9%)	37 (24.8%)	0.42	43 (28.9%)	43 (28.9%)	0.50	86 (57.7%)
Delivering laboratory samples (e.g., blood, urine, and stools) of patients	53 (35.3%)	32 (21.3%)	0.50	50 (33.3%)	35 (23.3%)	0.08	85 (56.7%)
Taking the patient to bed or transferring to another place	58 (38.9%)	37 (24.8%)	0.83	48 (32.2%)	47 (31.5%)	0.55	95 (63.8)
During clinic medical visit/academician visit	52 (34.7%)	35 (23.3%)	0.94	51 (34%)	36 (24%)	0.08	87 (58%)
Reasons for occupational accidents							
Intensive work tempo	64 (42.7%)	39 (26%)	0.42	48 (32%)	55 (36.7%)	0.02 *	103 (68.7%)
Lack of attention	60 (40%)	43 (28.7%)	0.45	51 (34%)	52 (34.7%)	0.25	103 (68.7%)
Not using PPE	61 (40.7%)	44 (29.3%)	0.46	48 (32%)	57 (38%)	0.009 **	105 (70%)
Long work hours	66 (44%)	47 (31.3%)	0.48	55 (36.7%)	58 (38.7%)	0.08	113 (75.3%)
Inexperience	60 (40%)	47 (31.3%)	0.12	53 (35.3%)	54 (36%)	0.22	107 (71.3%)
Notified person in case of hazardous incident							
I did not tell anyone.							
Yes	30 (20%)	21 (14%)	0.83	27 (18%)	24 (16%)	0.96	51 (34%)
I informed the supervising nurse, and he or she was interested.							
Yes	66 (44%)	45 (30%)	0.82	52 (34.7)	59 (39.3%)	0.01 *	111 (74%)
I informed the responsible academician, and he or she took an immediate interest.							
Yes	68 (45.3%)	47 (31.3%)	0.69	58 (38.7%)	57 (38%)	0.32	115 (76.7%)
The priorities when faced with a dangerous situation							
My priority is to comply with every precaution that our institution takes.							
Yes	69 (46%)	47 (31.3%)	0.81	55 (36.7%)	61 (40.7%)	0.01 *	116 (77.3%)
My own health is my top priority.							
Yes	76 (50.7%)	50 (33.3%)	0.85	63 (42%)	63 (42%)	0.13	126 (84%)
My priority is my job.							
Yes	54 (36%)	35 (23.3%)	0.83	44 (29.3%)	45 (30%)	0.33	89 (59.3%)
I have been given rest leave when I experienced clinical hazards.							
Yes	61 (40.7%)	36 (24%)	0.32	53 (35.3%)	44 (29.3%)	0.51	97 (64.7%)
When I encounter a dangerous situation, it is documented.							
Yes	67 (44.7%)	37 (24.7%)	0.09	53 (35.3%)	51 (34%)	0.52	104 (69.3%)

Note. NS: nursing student; OHS: occupational health and safety; PPE: personal protective equipment; * *p* < 0.05. ** *p* < 0.01.

## Data Availability

Data are not shared due to privacy and ethical restrictions.

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
