# Peer review of "Factors Influencing Occupational Health and Safety among Nursing Students in Their Clinical Placements: A Cross-Sectional Study"

_healthcare, 2024, doi:10.3390/healthcare12090884_

Round 1

Reviewer 1 Report

Comments and Suggestions for Authors

The manuscript entitled “Factors Influencing Occupational Health and Safety among Nursing 2 Students in Their Clinical Placements: A Cross-Sectional Study”. It can be suitable for printing with the following corrections.

There are a few comments:

  1. Choose the keywords based on the mesh and the link below.

       https://www.ncbi.nlm.nih.gov/mesh/

  1. It is better to mention the risk factors of occupational health and safety such as physical, mental, and social issues among nurses in the introduction.
  2. Cronbach’s α value in line 106 is not correct and the values 0.82 and 0.79 seem to be correct.
  3. The title of Table 1 is repeated.

- When the information is presented in the form of a table, there is no need to re-present all the information in writing outside the table 1.-

- The information in Table 1 can be presented in 6 columns to occupy less space.

  1. The type of statistical test used should be presented in the title of Table 2, 3 and 4.
  1. Due to the availability of information in the table, the explanation of tables should be reduced as much as possible.
  1. HCW in line 239 What words are abbreviations?
  1. No explanations have been presented about the significant difference between latex from chemicals between women and men in 4.1. Identifying threats section.
  • Other statistically significant items should also be mentioned in the discussion in related section and present reasons

Sincerely 

Reviewer 2 Report

Comments and Suggestions for Authors

Dear Authors!

This paper addresses a relevant topic and has a number of strengths. However, it does not yet meet the requirements for a high-quality paper in a top journal. There are a number of issues to be reconsidered and completed.

1. Materials and methods section: How did you ensure that all those who filled out the online questionnaires were nursing students and not other students or other people? Please describe.

2. The questionnaire was one adapted from a Turkish setting. What about the questionnaire translation and validation? Was this study the validation study of the translation? It seems so, but there are no methodological deliberations on this fact. 

3. Why is precisely semester 6 the cutoff point for lower and higher academic level? Please address. (Maybe the curriculum entails more hospital practice starting from the 7th semester?) Otherwise the academic year could be used as a continuous variable without grouping them into 2 groups. 

4. Most importantly: the Discussion only targets results of descriptive statistics. The valuable information to be found in Tables 2,3,4 and 5 are not addressed in this part, which is a big deficiency of the paper. Secondly, the structure of the Results should be structured in a similar way as the Discussion part, that is, following the same line of thought or subcategories (the four subcategories).

Further minor issues:

1. In line 134 it is stated: "are working and studying part-time". Is part-time studying possible in the country? What does this imply with respect to the results? Or is it only misphrasing? 

2. The Individual characteristics (from line 128) are not results, they should be placed to the Materials and methods section.

3. Table 2, subtitle: Findings and Symptoms in clinical practice - what do you mean by findings?

4. Line 155: PPE is used without previously being referred to, please use it in its complete form first. 

5. The subtitle starting from line 224 should not be a separate subchapter, as it is only 3 lines long. 

6. The word academic rank is quite unnown, I suggest to replace it with the year of study.

7. Subtitle 2.5 Ethical approval, the Declaration of Helsinki should be completed with some clarification like the World Medical Association's ethical principles.

8. Using the abbreviation NS to shorten nursing students is quite unnecessary. 

9. Subtitle 2.3, line 84: there is a mistyping in the title.

10. Line 41: more than half of the students were aware - which students you mean? Here other research is referred to.

11. Line 45: reference in brackets is missing about Quaraman.

12. The abbreviation OHS appears without being introduced, please complete. 

Please make the effort to improve the soundness and logical coherence of the paper. Good luck!

Author Response

Please the attachment

Reviewer 3 Report

Comments and Suggestions for Authors

It would be interesting to explain the implications of the results for the training of students and for the clinical supervision of students. What other research questions does this all this pose?

Round 2

Reviewer 2 Report

Comments and Suggestions for Authors

Dear Authors!
The paper has been significantly improved. Particularly its methodological soundness and the quality of the Discussion part have been much better elaborated. Congratulations for this considerable effort investment!

Best, reviewer